# Accurate computation of ionic concentrations in the synaptic cleft requires the full Poisson–Nernst–Planck (PNP) equations

Karoline Horgmo Jæger[ORCID]*, Aslak Tveito

Department of Computational Physiology, Simula Research Laboratory, Oslo, Norway

* karolihj@simula.no

## Abstract

The synaptic cleft between neighboring neurons is the site of neurotransmitter-mediated communication that underlies normal brain function, including learning and memory. When an action potential reaches the presynaptic terminal, released neurotransmitters cross the cleft under the combined influence of diffusion and electrical forces to activate postsynaptic receptors. Despite this, synaptic-cleft transport is commonly modeled using a pure diffusion model, neglecting electrical drift. Here, we quantify the relative contributions of diffusion and electrical terms in the Poisson–Nernst–Planck (PNP) framework and assess whether the pure diffusion approximation is adequate. We solve the full PNP system in a three-dimensional computational model of the synaptic cleft at nanometer-scale resolution, tracking five ionic species ($Na^+$, $K^+$, $Ca^{2+}$, $Cl^-$, $Glu^-$) with full spatial and temporal detail. Solutions are compared directly with those of the pure diffusion (D) model. The D and PNP models produce markedly different ionic concentration fields. Analysis of ionic fluxes confirms that diffusive and electrical contributions are of comparable magnitude across all species. These discrepancies are robust across parameter variations, including the number of AMPA receptors, the amount of released glutamate, the cleft height, and the cleft diffusion coefficient, and are amplified as the number of AMPA receptors or released glutamate ions increases, when the cleft becomes narrower or when diffusion becomes more restricted. However, because of competing effects, the resulting difference in the associated AMPA current is modest. The quantitative and qualitative differences between the pure D model and the full PNP model demonstrate that neglecting electrical forces in the synaptic cleft has consequences. These discrepancies are large enough to alter the predicted dynamics and biological interpretation of synaptic transmission, establishing that accurate computation of ionic concentrations in the synaptic cleft requires the full PNP equations.

**Data availability statement:** There are no primary data in the paper. The code used in our simulations are publicly available at Github (https://github.com/karolihj/PNP-synapse-FDM-2026).

**Funding:** KJ and AT were supported by the Research Council of Norway via grant #355113 (SCALES), grant #322312 (SIMBER), and grant #360005 (DigiCells), in addition to the SUURPh program funded by the Norwegian Ministry of Education and Research. The funders played no role in the study design, data collection and analysis, decision to publish, or preparation of the manuscript.

**Competing interests:** The authors have declared that no competing interests exist.

## Author summary

The synaptic cleft is the narrow gap between two communicating neurons. When an electrical signal arrives, the neurotransmitter glutamate is released into this gap and transported across to activate receptors on the postsynaptic side. The gap is only about 15–25 nm wide, making it very difficult to study experimentally, and mathematical models are therefore used to investigate the ion dynamics within it. The simplest approach models ion transport by pure diffusion. However, ions are charged particles and are subject to both diffusive and electrical forces. Including both leads to the Poisson–Nernst–Planck (PNP) equations, which are more challenging to solve computationally. We compare solutions of the full PNP equations with those of the pure diffusion model for the synaptic cleft and find that the differences are substantial. Our conclusion is that accurate computation of ionic concentrations in the synaptic cleft requires the full PNP equations.

## 1. Introduction

A large number of physiological processes depend on the concentrations of a few ionic species. The spatiotemporal dynamics of these concentrations have therefore been studied for decades, and a broad range of mathematical models have been developed. At the most fundamental level, ion transport is governed by diffusion and electrical drift. In small volumes, such as the dyadic cleft of cardiomyocytes and the synaptic cleft between two neighboring neurons, a common modeling strategy is to use compartment models with spatially averaged concentrations (constant in space within each compartment), see, e.g., [1,2]. At the other end of the spectrum, particle-based approaches explicitly track individual molecules or ions, often using Monte Carlo or random-walk descriptions, see, e.g., [3,4]. Within a continuum description, diffusion-only models based on Fick's law have frequently been used in synaptic and extracellular contexts, see, e.g., [5–7]. If both electrical and diffusional effects are included at the continuum level, the Poisson–Nernst–Planck (PNP) system must be used, see [8–14], or one may consider electroneutral (or quasi-electroneutral) reductions, see [15–17].

Here, we consider electrodiffusion in the synaptic cleft situated between a presynaptic and a postsynaptic neuron for a glutamatergic AMPA synapse. When the presynaptic membrane depolarizes, voltage-gated calcium channels open and the intracellular calcium concentration increases, leading to vesicular release of glutamate. Glutamate then crosses the synaptic cleft under the combined influence of diffusion and electrical forces. At the postsynaptic membrane, this activates AMPA receptors, allowing sodium and potassium ions to flow across the membrane. This transmission process underlies neuronal communication and is fundamental for normal brain activity, see, e.g., [18].

Neurotransmitter dynamics in the synaptic cleft has often been modeled using diffusion-only descriptions, see, e.g., [7,19], and impressive analytical results have

been obtained within this framework. However, when electrical forces are included, analytical solutions are generally not available [20]. Moreover, the full PNP system is regarded as numerically challenging because it is strongly coupled and nonlinear, and because thin Debye layers near membranes introduce steep spatial gradients and stiffness [11,21]. By using the fully implicit scheme introduced in [14,22], we can solve the full PNP equations in the synaptic cleft geometry and thereby evaluate the quantitative effect of including, or omitting, the electrical drift terms. To the best of our knowledge, the full PNP system has not previously been applied to the synaptic cleft. The most closely related prior work [23] analyzed the effect of the postsynaptic current on neurotransmitter diffusion using a single Nernst–Planck equation for the neurotransmitter alone, with an externally imposed electric field. By contrast, the full PNP system tracks all ionic concentrations self-consistently and couples them with the electric potential.

## 2. Results

We compare solutions of the full PNP model to the solutions of the pure diffusion (D) model for synaptic dynamics following synaptic vesicle opening, to assess whether the D model is sufficient or whether the full PNP equations are required. The considered computational domain is illustrated in Fig 1, and the equations defining the PNP and D models are described in the Methods section.

### 2.1. Synapse dynamics

Before turning to the comparison of the PNP and D model solutions, Fig 2 gives an overview of the dynamics considered in these simulations, i.e., the dynamics in the extracellular cleft between two neurons in an excitatory synapse. The displayed dynamics are taken from a PNP model simulation. Before the simulation starts, a propagating action potential is assumed to have traveled through the axon of the presynaptic cell and have reached the membrane of the presynaptic cell that is facing the synaptic cleft. Moreover, this depolarization has activated $Ca^{2+}$ channels in the presynaptic membrane, and the increased intracellular $Ca^{2+}$ concentration has triggering synaptic vesicles to dock at and fuse with the presynaptic membrane. This allows the neurotransmitter glutamate to diffuse out of the vesicle into the synaptic cleft. Our

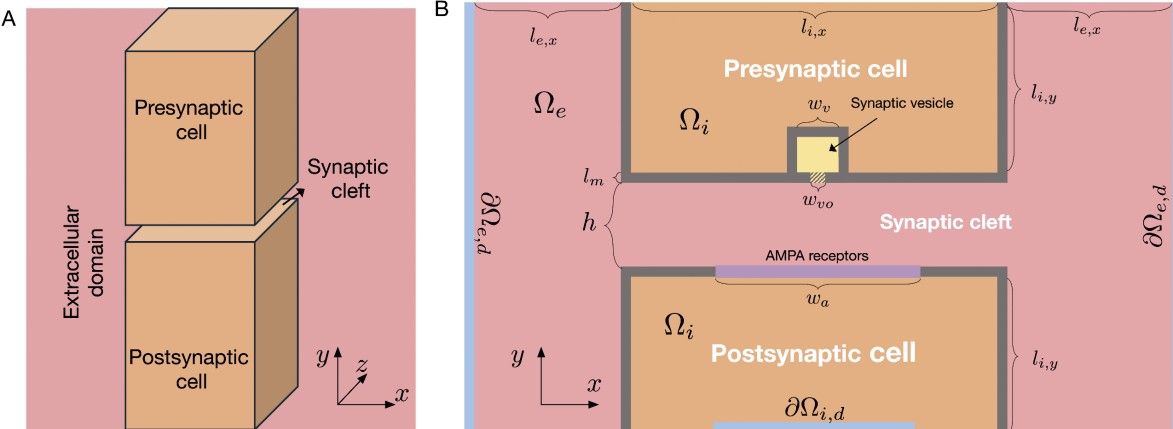

**Fig 1. Illustration of the computational domain.** A: Overview of the 3D domain. We consider a small part of a presynaptic cell and a small part of a postsynaptic cell, surrounded by an extracellular domain. B: More detailed view of the domain components in a sheet in the center of the domain in the $x$- and $y$- directions. In the membrane of the presynaptic cell, a synaptic vesicle is located near the membrane with a possible opening to the synaptic cleft. The width of this opening is denoted by $w_{vo}$. In addition, AMPA receptors are embedded in the membrane of the postsynaptic cell. The height of the synaptic cleft, i.e., the distance between the presynaptic and postsynaptic membranes, is denoted by **$h$**. The parts of the domain boundary denoted by $\partial\Omega_{e,d}$ and $\partial\Omega_{i,d}$, are parts of the boundary where Dirichlet boundary conditions are applied. Neumann boundary conditions are applied in the remaining parts of the domain boundary. The domain lengths are provided in Table 1.

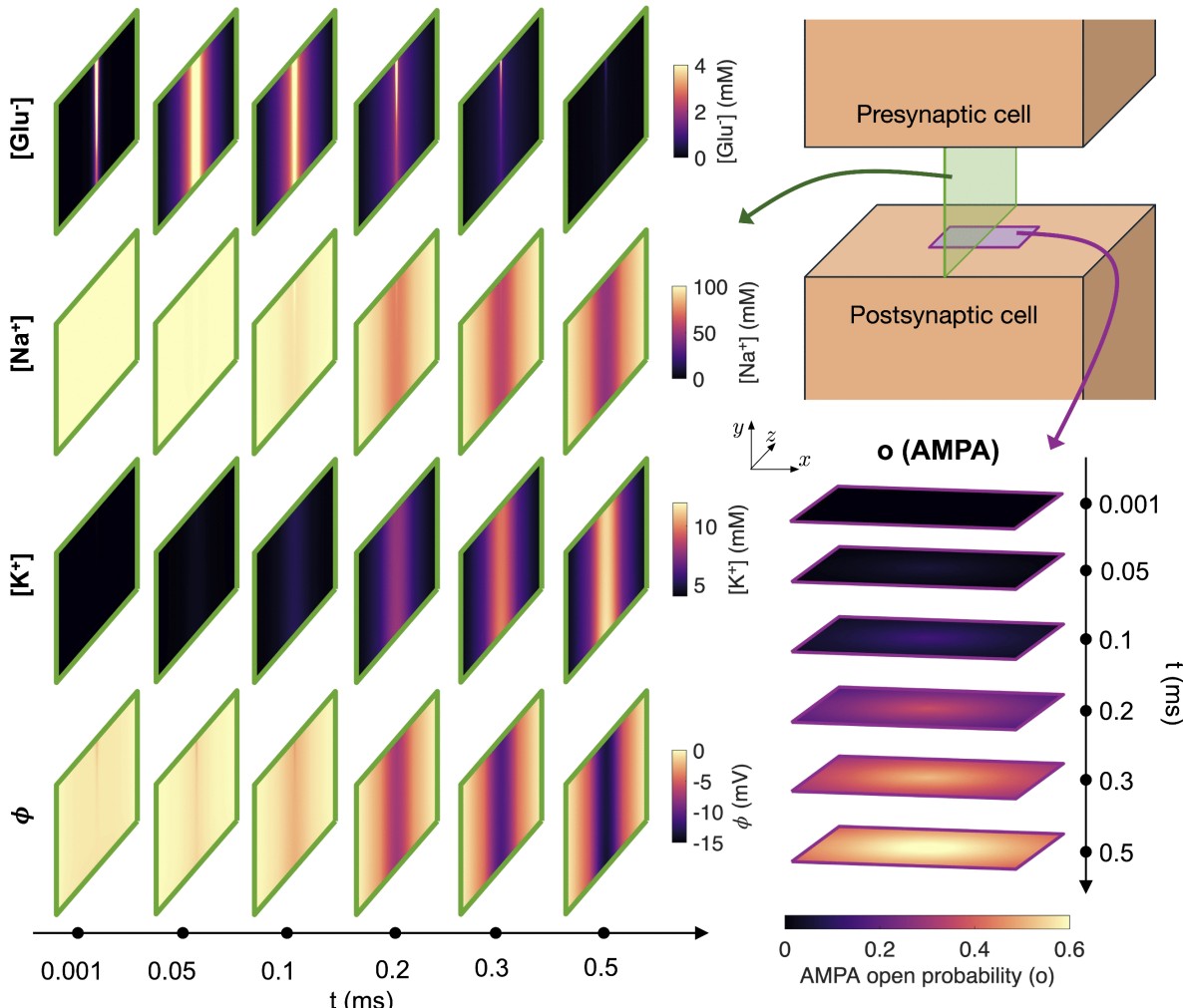

**Fig 2. Dyad dynamics following synaptic vesicle opening in a PNP model simulation.** The concentration of Glu$^-$, Na$^+$ and K$^+$ and the electrical potential, $\phi$, are displayed in a sheet in the y- and z-directions in the center of the domain at six different points in time. In addition, the AMPA open probability is displayed in the sheet defining the AMPA receptor zone of the postsynaptic membrane (see Fig 1B). Note that the y- and z-axes are not shown with equal physical scaling; the y-direction, corresponding to the cleft height, is enlarged relative to the z-direction to make the narrow cleft visible. The displayed domain is actually about 40 times longer in the z-direction than in the y-direction. In S1 Fig in the Supporting Information, the glutamate concentration is displayed with equal physical axes.

simulations begin at this release. In Fig 2, we observe that after 0.001 ms and 0.05 ms (i.e., at 1 $\mu$s and 50 $\mu$s), the glutamate concentration in the cleft increases, in particular near the synaptic vesicle opening in the center.

After a while, at 0.1 ms, the increased glutamate concentration outside the AMPA receptors cause the AMPA receptor open probability to increase slightly (see rightmost panel). When open, the AMPA receptors allow for Na$^+$ to flow from the extracellular space (where the concentration initially is 100 mM) to the intracellular space of the postsynaptic cell (where the concentration initially is 12 mM). In addition, K$^+$ is allowed to flow through the open AMPA receptors from the intracellular space of the postsynaptic cell (where the concentration initially is 125 mM) to the synaptic cleft (where the concentration initially is 4 mM). Since the intracellular potential of the postsynaptic cell is set to a holding potential of −70 mV, which is close to $v_{0,K^+}$ and far from $v_{0,Na^+}$, more Na$^+$ flows into the presynaptic cell than K$^+$ flows out of the cell (see (9) and (10))

resulting in a depolarizing transmembrane current that makes the postsynaptic cell more likely to fire an action potential. Thus, an electrical signal has been sent from the presynaptic cell to the postsynaptic cell.

In Fig 2, we observe that at $t = 0.2$ ms, 0.3 ms and 0.5 ms the open probability of the AMPA receptors increases leading to increased AMPA receptor transmembrane current. As more Na$^+$ flows into the postsynaptic cell, the local Na$^+$ concentration in the center of the synaptic cleft decreases, and as more K$^+$ flows out of the postsynaptic cell, the local K$^+$ concentration in the center of the cleft increases.

In the lower panel of Fig 2, the electrical potential, $\phi$, is displayed at the considered time steps. We observe that the synaptic vesicle opening causes a small non-zero electrical potential close to the vesicle opening. Furthermore, as the AMPA receptors open, a substantial electrical potential is present in the cleft. In the full PNP model, the observed potential gradients could affect the dynamics of the ionic species in the cleft (see (6)), whereas in the D model, the effect of the electrical field is neglected (see (8)). The purpose of this study is to investigate whether the D model neglecting the effect of the electrical field is sufficiently accurate to capture the dynamic of the synaptic cleft, or whether the electrical effects are significant and the full PNP model is required.

## 2.2. Comparison of PNP and D solutions

We now move on to compare the solutions of the PNP and D models. Fig 3 shows ionic concentration solutions from PNP and D simulations using the default simulation setup, along with the electrical potential computed in the PNP model simulation. The electrical potential is not a part of the D model solution (see (8)). We focus on the time point 0.5 ms after synaptic vesicle opening, except in the upper right panel, which shows the solutions in one spatial point as functions of time.

Focusing first on the glutamate concentration, we observe that initially [Glu$^-$] is zero in the cleft, but it increases rapidly when the synaptic vesicle is opened. Next, the concentration decreases rapidly as the ions flow out of the synaptic cleft. However, the time course of the changes in glutamate concentration appears to be somewhat different in the PNP and D solutions. Indeed, in spatial sheet and line plots of the glutamate concentration at $t = 0.5$ ms, we observe that the concentration is about 50% higher in the D solution compared to the PNP solution in the center of the cleft.

Na$^+$ ions flow through the AMPA receptors in the direction from the synaptic cleft to the postsynaptic intracellular space. In Fig 3, we observe that this results in a local Na$^+$ depletion in the center of the synaptic cleft (as also observed in Fig 2). However, the magnitude of the changes in Na$^+$ concentration is markedly different in the PNP and D cases. Similarly, K$^+$ ions flow through the AMPA receptors in the direction from the postsynaptic cell to the synaptic cleft. This leads to a locally increased K$^+$ concentration in the center of the cleft, and there is a substantial difference in the size of this increase between the PNP and D cases.

The AMPA receptors are not assumed to carry Ca$^{2+}$ or Cl$^-$ ions between the intracellular and extracellular spaces. Therefore, the concentrations of these ions do not change with time in the D model simulation. In the PNP model, on the other hand, these ions are affected by electrical drift and we observe considerable local changes in the Ca$^{2+}$ and Cl$^-$ concentrations in the cleft center in the PNP model solution.

## 2.3. Ionic fluxes

In order to examine the cause of the large differences observed between the PNP and D model solutions in Fig 3, we consider the following fluxes governing the flow of ions:

$$J_d = -D_k \nabla [k],$$

(1)

$$J_e = -\frac{D_k z_k e}{k_B T}[k]\nabla\phi.$$

(2)

PLOS Computational Biology

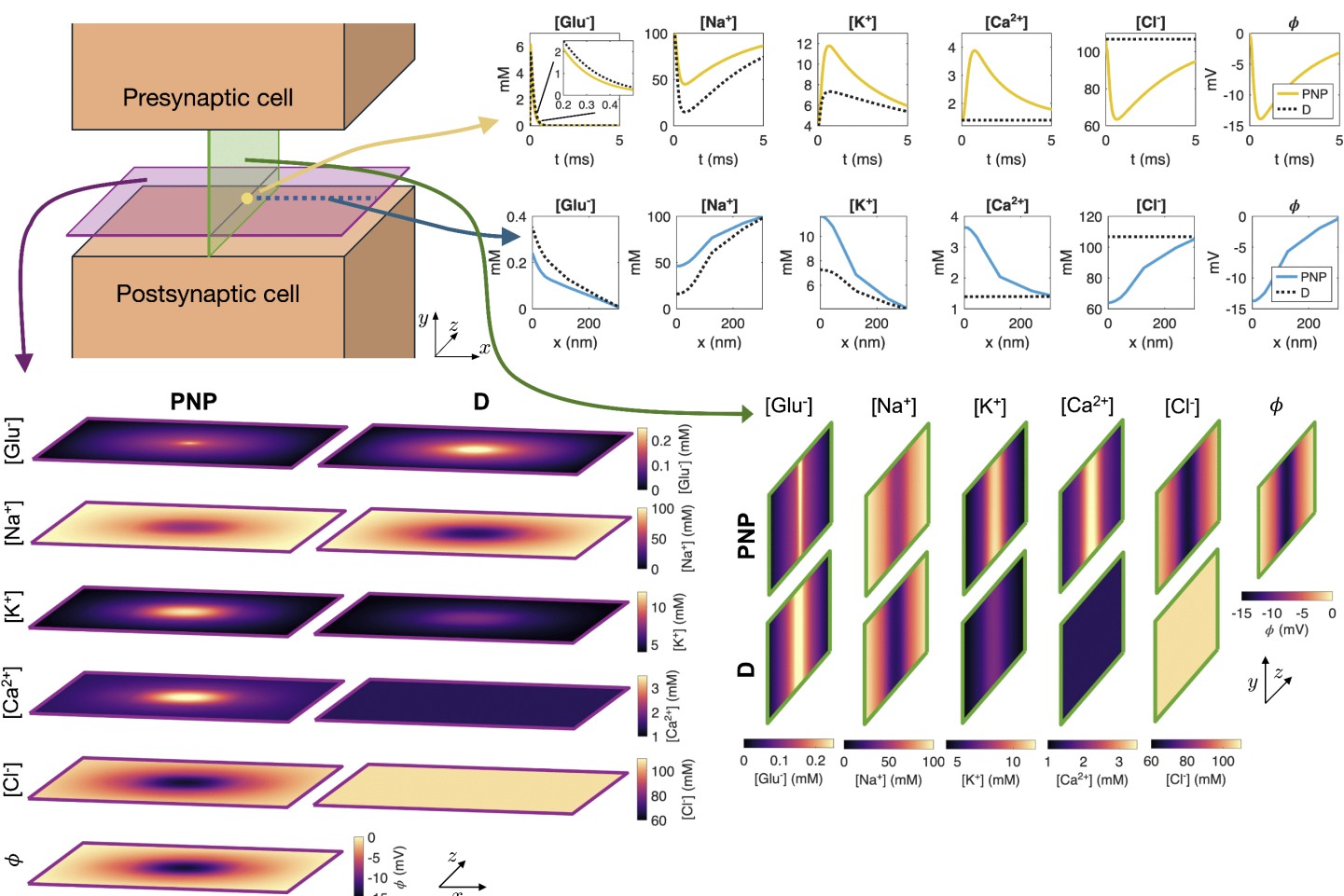

**Fig 3. Solution of the PNP model and the pure diffusion model (D).** The ionic concentrations are displayed for both models as well as the electrical potential, $\phi$, for the PNP model. In the lower left panel, solutions at $t = 0.5$ ms are displayed in a sheet in the $x$- and $z$-directions close to the postsynaptic membrane. In the lower right panel, solutions at $t = 0.5$ ms are shown in a sheet in the $y$- and $z$-directions in the center of the domain. In the upper panel, the solutions in a point in the center of the domain in the $x$- and $z$-directions, close to the postsynaptic membrane in the $z$-direction are plotted as functions of time. Below, the solutions at $t = 0.5$ ms are plotted along a line in the $x$-direction close to the postsynaptic membrane. Here, $x = 0$ nm marks the center of the domain in the $x$-direction. Note, again, that the $x$-, $y$- and $z$-axes are not shown with equal physical scaling; the $y$-direction, corresponding to the cleft height, is enlarged relative to the $x$- and $z$-directions to make the narrow cleft visible. The displayed domain is actually about 40 times longer in the $x$- and $z$-directions than in the $y$-direction.

We refer to $J_d$ as the diffusional flux and $J_e$ as the electrical flux. In terms of fluxes, the ion concentrations are governed by

$$\frac{\partial[k]}{\partial t} = -\nabla \cdot (J_d + J_e)$$

(3)

in the PNP model and by

$$\frac{\partial[k]}{\partial t} = -\nabla \cdot J_d$$

(4)

in the D model.

In Fig 4, we examine the *x*-component of the two fluxes computed from the PNP model solution. In the upper panel, the fluxes at $t = 0.5$ ms are plotted along a line in the *x*-direction, and in the lower panel the fluxes in a point 60 nm to the right of the center of the synaptic cleft are plotted as functions of time. We observe that for Na$^+$, the diffusional flux acts to move Na$^+$ ions into the cleft center (because of the local Na$^+$ depletion). In addition, the electrical flux also acts to move Na$^+$ ions toward the cleft center due to the direction of the local electrical field in the cleft. The total flux driving Na$^+$ towards the cleft center, $J_d + J_e$, considered in the PNP model is therefore greater than the diffusion flux, $J_d$, considered in the D model, alone. This can explain why the Na$^+$ concentration is larger in the cleft center for the PNP model compared to the D model (see Fig 3).

For the K$^+$ concentration, the diffusional flux acts to move K$^+$ ions out of the cleft because of the local increased K$^+$ concentration caused by the AMPA current. However, the electrical flux acts to move K$^+$ into the cleft because of the direction

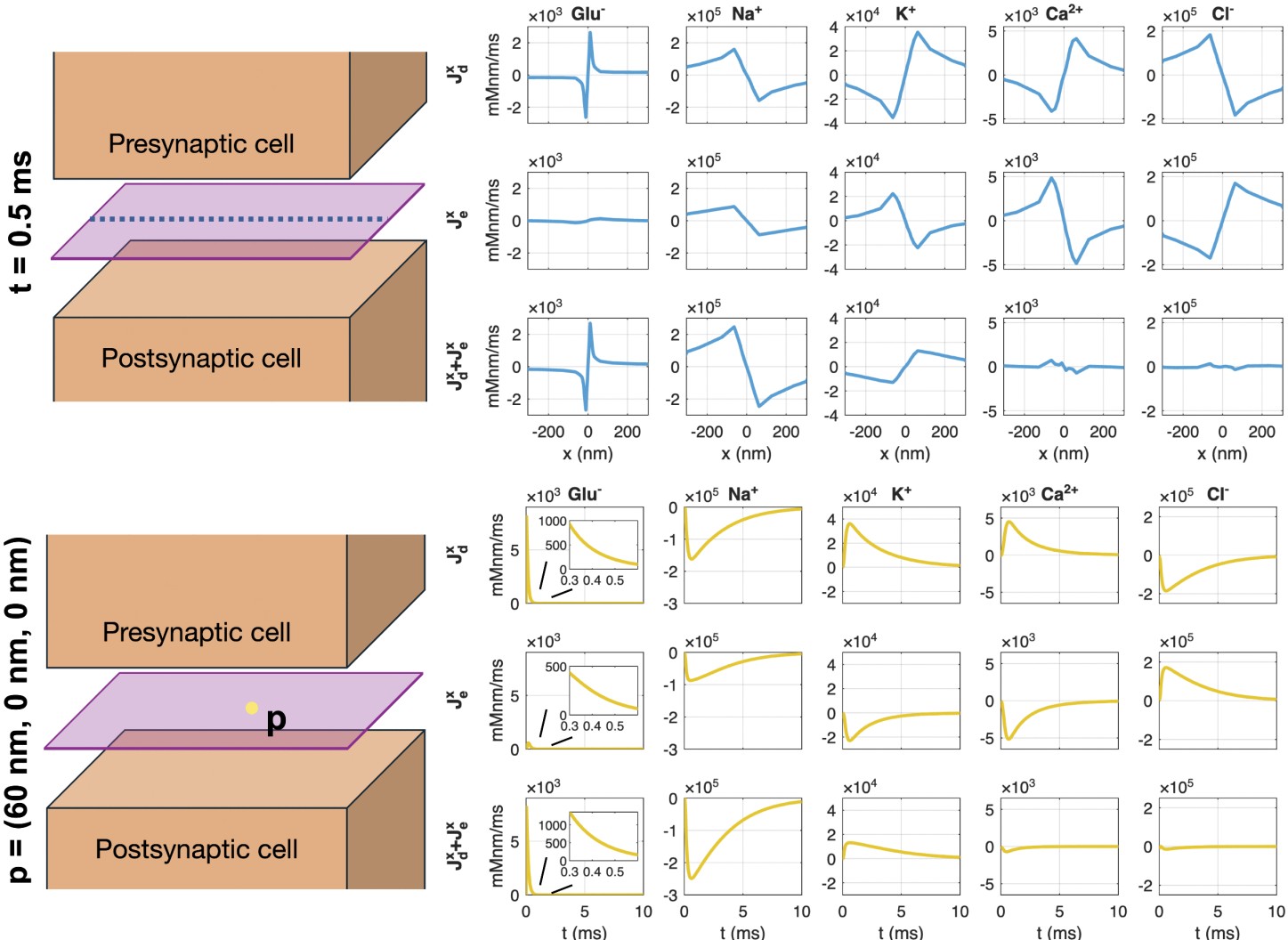

**Fig 4. Diffusional and electrical fluxes.** The *x*-component of the diffusional ($J_d^x$) and electrical ($J_e^x$) fluxes is shown for each ion species. The fluxes are computed from the solution of the PNP model. The upper panel shows the fluxes as functions of *x* along the center of the domain in the *y*- and *z*-directions at $t = 0.5$ ms, and the lower panel shows the fluxes as functions of time in a point 60 nm to the left of the center of the domain. For glutamate, the insets in the lower panel shows the fluxes in the time interval from 0.3 ms to 0.6 ms.

of the electrical field. Therefore, the total flux, $J_d + J_e$, included in the PNP model is smaller than the diffusion flux, $J_d$, alone, and, consequently, the K$^+$ concentration in the cleft center is larger for the PNP model than for the D model.

For glutamate, the diffusional flux acts to move ions out of the cleft due to the elevated concentration in the cleft center following the release of glutamate from the synaptic vesicle. Moreover, the electrical flux also acts to move Glu$^-$ ions out of the cleft center due to the direction of the electrical field and the negative charge of the ions. Therefore, the flux driving Glu$^-$ out of the cleft is larger in the PNP model (including both $J_d$ and $J_e$) compared to the D model (including only $J_d$), explaining why the Glu$^-$ concentration is smaller in the PNP model compared to the D model at $t = 0.5$ ms (see Fig 4).

For Ca$^{2+}$ and Cl$^-$, the electrical flux drives the ions towards and away from the cleft center, respectively, because of the positive and negative charge of the ions. This leads to a Ca$^{2+}$ increase and a Cl$^-$ decrease in the cleft center. The diffusional flux acts to counteract these effects. However, in the D model, $J_d$ is zero for these ions because the electrical flux does not create any concentration gradients (see Fig 3).

## 2.4. AMPA receptor current

In the left panel of Fig 5, we compare the total current through the AMPA receptors of the postsynaptic membrane, $I_{AMPA}$ (defined in (12)), computed using the PNP and D models. Overall, we observe that the peak current computed using the PNP model is approximately 5% more negative than in the D model. However, this net difference is the composite of several competing mechanisms, as investigated in the right panels of Fig 5.

Specifically, two effects work to make the peak $I_{AMPA}$ *less* negative for the PNP model than for the D model: (*i*) a reduced peak open probability occurs because the electrical force drives glutamate ions out of the cleft, reducing the local glutamate concentration in the PNP case; and (*ii*) a less negative membrane potential, $v = \phi_i - \phi_e$, is observed because the PNP model captures a negative extracellular potential in the cleft (assumed to be zero in the D model), resulting in a less negative $v$.

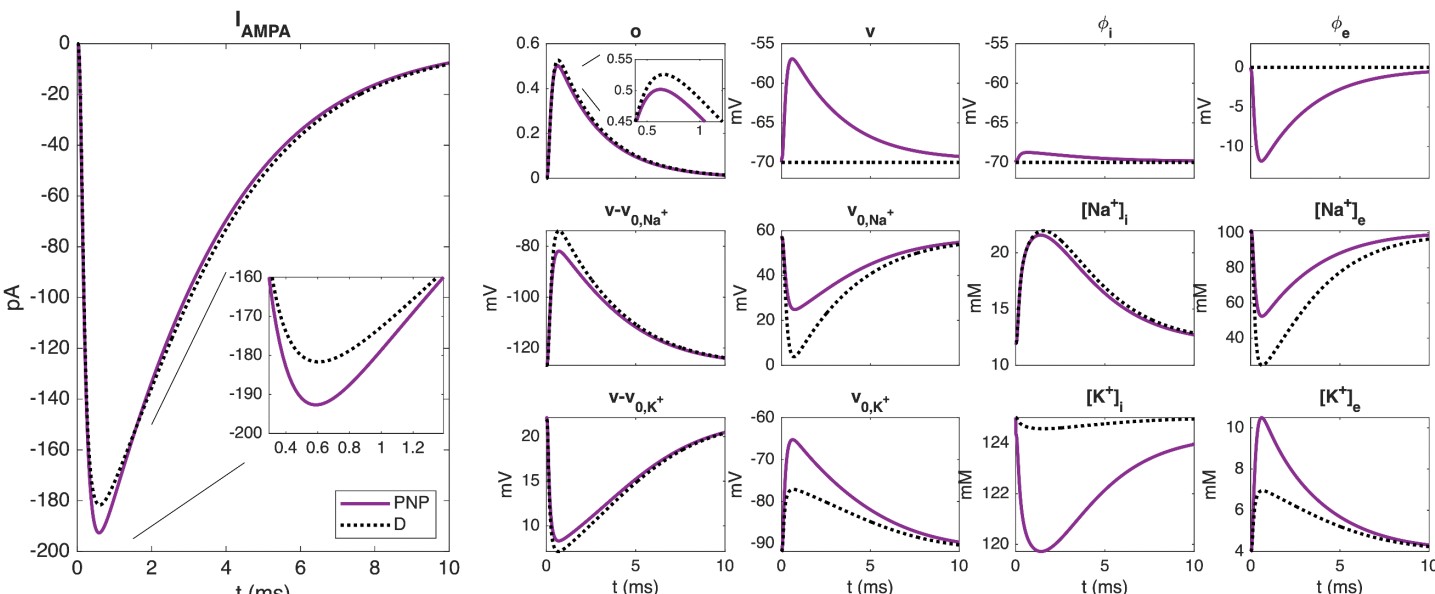

**Fig 5. Total AMPA receptor current and associated terms in the PNP and pure diffusion (D) models.** Solutions correspond to the simulations presented in Fig 3. The AMPA receptor current is integrated over the AMPA receptor area, while all other properties are spatially averaged over the same area. The net difference in AMPA current between the PNP and D models is the result of competing electrodiffusive effects.

Conversely, two effects work to make the peak $I_{AMPA}$ *more* negative for the PNP model than for the D model: (*i*) a more positive $v_{0,Na^+}$ arises because the electrical force moves Na$^+$ ions toward the cleft center, meaning the extracellular sodium concentration, [Na$^+$]$_e$, is less depleted in the PNP case. This makes $v_{0,Na^+}$ more positive, which in turn makes the driving force contribution $-v_{0,Na^+}$ more negative; and (*ii*) a less negative $v_{0,K^+}$ occurs because, similarly, the electrical force draws K$^+$ ions toward the cleft center, causing a greater increase in [K$^+$]$_e$ in the PNP case. This makes $v_{0,K^+}$ less negative, rendering the $-v_{0,K^+}$ term less positive.

The exact difference in AMPA receptor current between the PNP and D models observed in Fig 5 is the net balance of these opposing electrodiffusive effects.

## 2.5. Parameter variations

In the simulations reported above, we have used the default parameter values reported in Table 1. In Figs 6 and 7, we examine the difference between the PNP and D models when some of the model parameters are adjusted.

**Table 1. Parameter values used in the simulations. Here, $\Omega_e$ refers to the extracellular domain, $\Omega_m$ refers to the membrane domain, and $\Omega_i$ refers to the intracellular domain (see Fig 1). In the cell membrane ($\Omega_m$), the diffusion coefficients are set to zero for all ions, and in the synaptic cleft and the synaptic vesicle opening, the default diffusion coefficients are divided by a factor $\kappa$.**

| Parameter | Description | Value | Ref. |
|---|---|---|---|
| $F$ | Faraday's constant | 96485.3365 C/mol | [26] |
| $e$ | Elementary charge | $1.60217662 \cdot 10^{-19}$ C | [26] |
| $k_B$ | Boltzmann constant | $1.380649 \cdot 10^{-20}$ mJ/K | [26] |
| $T$ | Temperature | 310 K | |
| $\varepsilon_0$ | Vacuum permittivity | 8854 fF/m | [26] |
| $\varepsilon_1$ | Relative permittivity, $\varepsilon_r$, in $\Omega_e$ and $\Omega_i$ | 80 | [27] |
| $\varepsilon_m$ | Relative permittivity, $\varepsilon_r$, in $\Omega_m$ | 2 | [27] |
| $D_{Na^+}$ | Default diffusion coefficient for Na$^+$ | $1.33 \cdot 10^6$ nm$^2$/ms | [28] |
| $D_{K^+}$ | Default diffusion coefficient for K$^+$ | $1.96 \cdot 10^6$ nm$^2$/ms | [28] |
| $D_{Ca^{2+}}$ | Default diffusion coefficient for Ca$^{2+}$ | $0.71 \cdot 10^6$ nm$^2$/ms | [28] |
| $D_{Cl^-}$ | Default diffusion coefficient for Cl$^-$ | $2.03 \cdot 10^6$ nm$^2$/ms | [28] |
| $D_{Glu^-}$ | Default diffusion coefficient for Glu$^-$ | $0.86 \cdot 10^6$ nm$^2$/ms | [29] |
| $\kappa$ | Reduction factor for diffusion in the cleft | 2.56 | [29] |
| $z_{Na^+}$ | Valence of Na$^+$ | 1 | |
| $z_{K^+}$ | Valence of K$^+$ | 1 | |
| $z_{Ca^{2+}}$ | Valence of Ca$^{2+}$ | 2 | |
| $z_{Cl^-}$ | Valence of Cl$^-$ | $-1$ | |
| $z_{Glu^-}$ | Valence of Glu$^-$ | $-1$ | |
| $N_{AMPA}$ | Number of AMPA receptors | 200 | [19] |
| $g_{AMPA,Na^+}$ | Single channel AMPA receptor conductance for Na$^+$ | 25 pS | [19] |
| $g_{AMPA,K^+}$ | Single channel AMPA receptor conductance for K$^+$ | 15 pS | [19] |
| $h$ | Height of the synaptic cleft | 15 nm | [19] |
| $l_m$ | Membrane thickness | 5 nm | [30] |
| $w_a$ | Width of the AMPA receptor area | 140 nm | [19,31] |
| $w_v$ | Synaptic vesicle diameter | 40 nm | [7] |
| $w_{vo}$ | Width of opening between synaptic vesicle and cleft | 4 nm | |
| $l_{e,x}, l_{e,z}$ | Extracellular domain length in *x*-, and *z*- directions | 1000 nm | |
| $l_{i,x}, l_{i,z}$ | Total width of synaptic cleft | 600 nm | [19,31] |
| $l_{i,y}$ | Intracellular domain length in the *y*-direction | 1000 nm | |

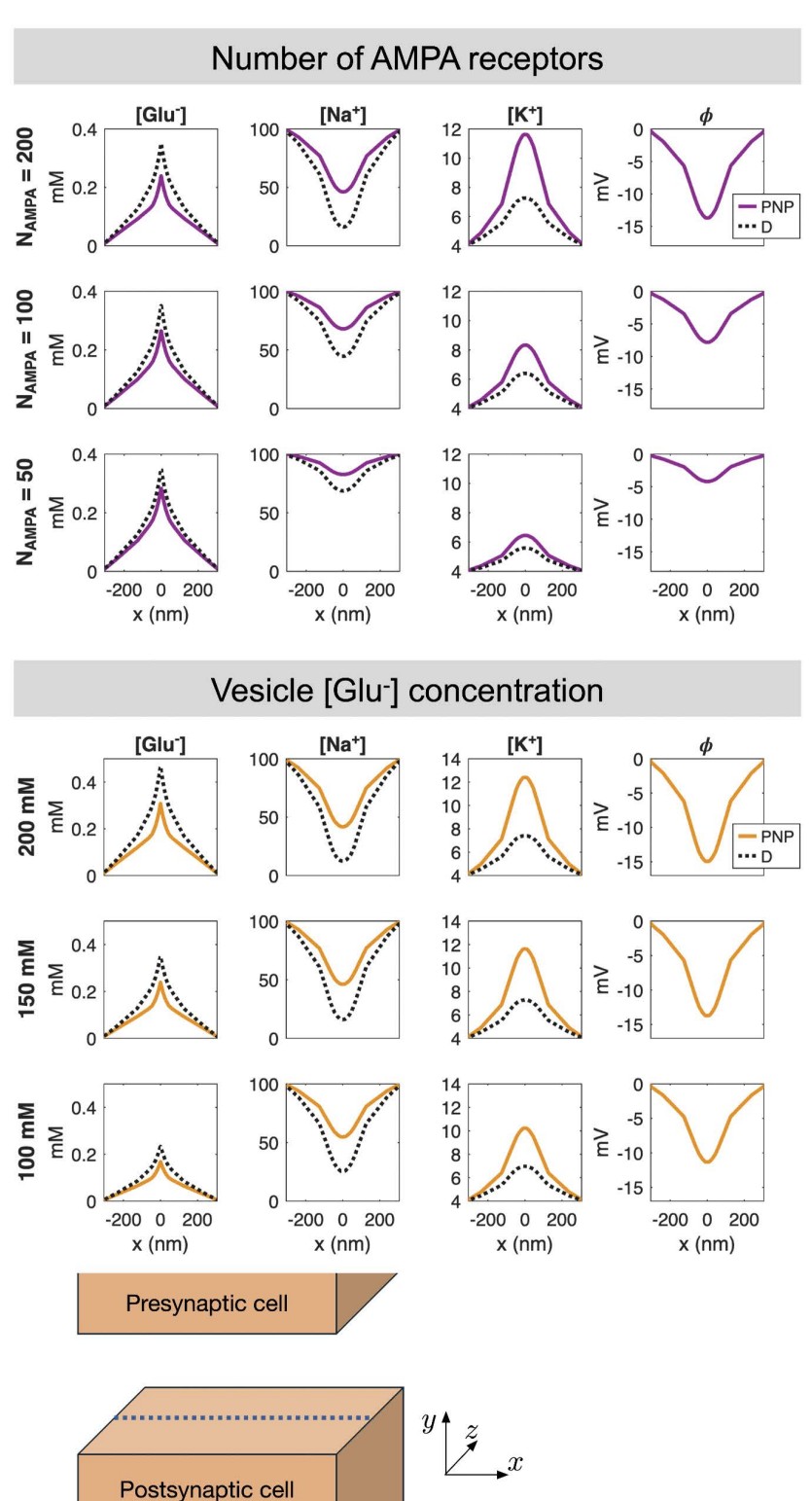

**Fig 6. Solutions of the PNP and D models for variations in the number of AMPA receptors and the vesicle [Glu⁻] concentration.** The concentrations of Glu⁻, Na⁺, K⁺ and $\phi$ at t = 0.5 ms are displayed along a line in the center of the synaptic cleft in the *y*- and *z*-directions. The potential is only

shown for PNP, because it is not a part of the D solution. The number of AMPA receptors and the vesicle [Glu⁻] concentrations applied in each case are reported to the left of each row. The remaining parameter values are as reported in Table 1. Note that when the vesicle [Glu⁻] concentration is adjusted the remaining vesicle concentrations are adjusted by the same factor to maintain initial electroneutrality (see Table 2).

**2.5.1. Varying the number of AMPA receptors.** In the upper panel of Fig 6, we consider three different values for the number of AMPA receptors on the postsynaptic membrane. We observe that as the number of AMPA receptors is increased, the [Na⁺] decrease and the [K⁺] increase resulting from AMPA receptor opening become more pronounced. In addition, the magnitude of the electrical potential in the cleft increases as the number of AMPA receptors increases. As expected, the more pronounced $\phi$ results in an increased difference in [Glu⁻] between the PNP and D solutions when the number of AMPA receptors increases. Specifically, the glutamate concentration is not affected by the number of AMPA receptors in the D model, but in the PNP model, electrical drift reduces [Glu⁻] more extensively as the number of AMPA receptors increases.

**2.5.2. Varying the number of released glutamate molecules.** Next, in the lower panel of Fig 6, we consider variations in the concentration of glutamate in the synaptic vesicle. We consider the concentrations 100 mM, 150 mM, and 200 mM, corresponding to approximately 4000, 6000, and 8000 glutamate molecules in the vesicle, respectively. When the number of released glutamate molecules increases, the AMPA receptor current seems to increase, leading to a larger electrical field in the cleft and larger differences between the PNP and D solutions.

**2.5.3. Varying the cleft height.** In the upper panel of Fig 7, we investigate the effect of the cleft height, $h$, i.e., the distance between the presynaptic and postsynaptic membranes. As $h$ decreases, the synaptic cleft becomes less spacious and fluxes into or out of the cleft have a larger effect on the concentrations. This is observed for all the ionic species considered in the upper panel of Fig 7. In addition, the magnitude of the extracellular potential in the cleft increases as the cleft height is reduced, leading to larger electrical drift and larger differences between the PNP and D solutions.

**2.5.4. Varying the cleft diffusion coefficients.** Finally, in the lower panel of Fig 7, we vary the value of the factor, $\kappa$, by which the default diffusion coefficients in Table 1 are divided in the synaptic cleft. We observe that increasing $\kappa$, i.e., reducing the diffusion coefficients in the cleft, has a similar effect as that observed for reducing the cleft height. By reducing the ease with which the ions may flow, the fluxes into or out of the cleft cause larger changes in the concentrations and a larger electric field. This results in larger differences between the PNP and D solutions.

## 3. Discussion

The synaptic cleft is the site of neurotransmitter-mediated signaling that underlies normal brain function. The ionic concentration dynamics in the cleft determine how, and how efficiently, neurotransmitters activate postsynaptic receptors. This makes accurate mathematical modeling of transport in the cleft important. Our results indicate that the full Poisson–Nernst–Planck system is required for this purpose, and we discuss this finding in the following.

### 3.1. A simple 1D problem illustrating that D is different from PNP

Our aim is to clarify whether the dynamics in the synaptic cleft can be adequately captured by the pure diffusion model (D), see (8), or whether the full Poisson–Nernst–Planck (PNP) model, see (5)–(7), must be solved. The appeal of D is its computational simplicity and, in some cases, analytical tractability. The full PNP system is more expensive because it is nonlinear and strongly coupled. In our simulations, however, the D and PNP solutions differ substantially, indicating that omitting electrical drift can change the predicted concentration dynamics.

A simple illustration can be obtained in one spatial dimension. Consider the interval (0,10) nm with homogeneous Neumann (zero-flux) boundary conditions for all concentrations. Let the initial conditions be the electroneutral mixture

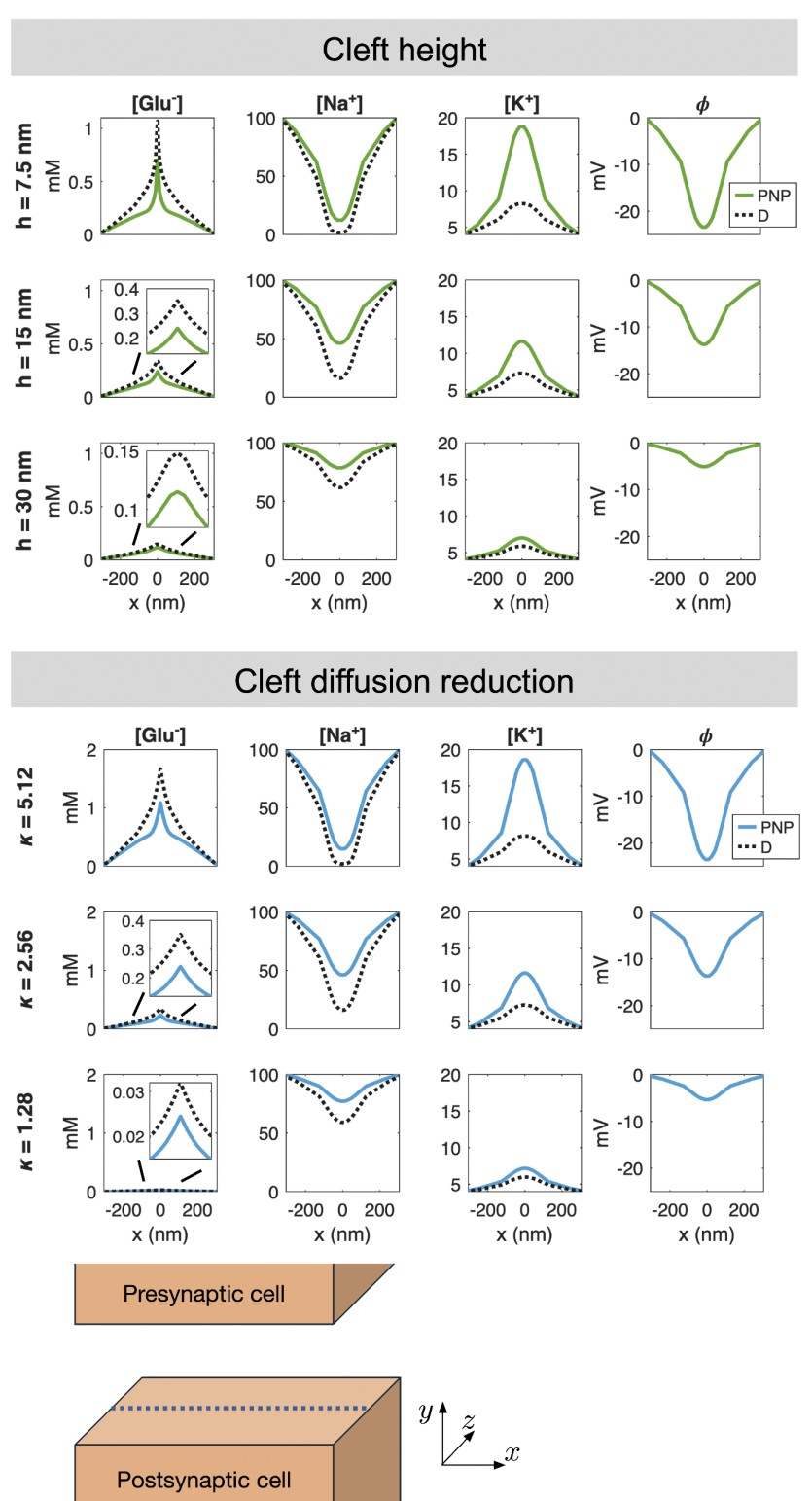

**Fig 7. Solutions of the PNP and D models for variations in the cell distance and the diffusion coefficient reduction factors in the cleft.** The concentrations of Glu⁻, Na⁺, K⁺ and $\phi$ at $t = 0.5$ ms are displayed along a line in the center of the synaptic cleft in the $y$- and $z$-directions. The potential is only shown for PNP, because it is not a part of the D solution. The cell distance, $h$, and the diffusion reduction factor, $\kappa$, applied in each case are reported to the left of each row. The remaining parameter values are as reported in Table 1.

[Na$^+$] = 100 mM, [Cl$^-$] = 99 mM, and [Glu$^-$] = 1 mM. For the PNP model, we additionally impose Dirichlet boundary conditions for the electrical potential: $\phi(0) = 0$ and $\phi(10\,\text{nm}) = \phi_0$, where $\phi_0$ is set to −5 mV, −10 mV, or −15 mV. In this setup, the D solution remains constant, whereas the PNP solution develops pronounced spatial variations driven by the boundary conditions of the electrical potential. Stationary solutions for this problem are shown in Fig 8. We also note that the deviation between the D and PNP solutions increases as the imposed electrical field becomes stronger (i.e., as $|\phi_0|$ increases).

### 3.2. Electrical drift contributes significantly to the ionic dynamics of the synaptic cleft

The results shown in Fig 3 and Figs 5–7 consistently show that the electrical term in the PNP equations has a substantial effect on the solution. In absolute terms, the perturbation is largest for ions present at high concentrations, but in relative terms all ionic species are significantly affected. This observation is robust across the parameter variations considered: it holds when the number of AMPA receptors, the initial glutamate concentration, the cleft height, and the ionic diffusion coefficients are varied. In all cases, the differences between the pure diffusion model (8) and the full PNP model (5)–(6) are substantial. Fig 4 provides a mechanistic explanation: the diffusive and electrical fluxes are of comparable magnitude for all ionic species. Retaining diffusion while discarding electrical drift therefore amounts to neglecting a contribution of similar size, which cannot be expected to yield an accurate approximation. Interestingly, these substantial differences in ionic concentrations do not translate into equally large differences in the resulting AMPA current, because the electrodiffusive effects entering the current act partly in opposite directions and therefore partially cancel; see Fig 5.

### 3.3. Computational cost

The full PNP model is generally regarded as computationally demanding relative to the pure D model. This depends, however, on the implementation, and efficient solvers for coupled systems of partial differential equations are an active area of research. We do not claim optimal solvers for either D or PNP, but it is useful to report the actual computing times observed in our simulations. The discretization is described in detail in [14], and for a 5 ms simulation the D code used about six minutes whereas the PNP code used about 400 minutes. Both used a time step of 0.02 ms. Note that earlier, we had to use a time step that was a million times shorter for the PNP models, see [13]. The fully coupled scheme developed in [14,22] has enabled reasonable CPU times, although the D model is still 67 times faster.

### 3.4. Limitations

The geometry used here is idealized — a flat cleft with parallel pre- and postsynaptic membranes — primarily to isolate the effect of the governing equations from confounding geometric factors. The actual geometry of the synaptic cleft is more complex, and morphological details such as membrane curvature and cleft width variations are known to influence

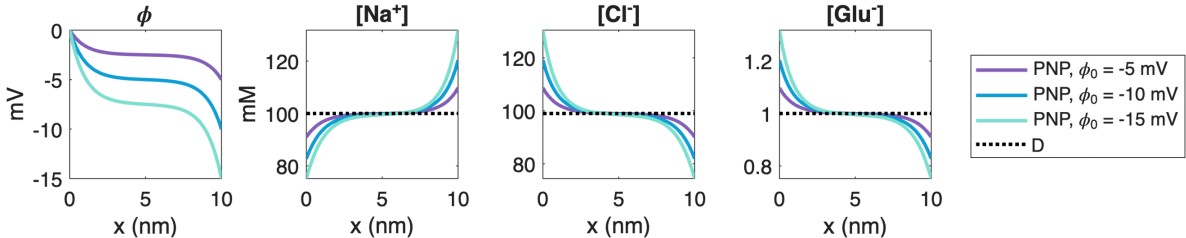

**Fig 8. Stationary solutions of a simplified one-dimensional problem.** We consider three ion species, Na$^+$, Cl$^-$, and Glu$^-$, with initial conditions [Na$^+$] = 100 mM, [Cl$^-$]= 99 mM, and [Glu$^-$] = 1 mM. We apply Dirichlet boundary conditions of $\phi = 0$ mV at $x = 0$ nm and $\phi = \phi_0 = $ −5 mV, −10 mV, or −15 mV for the electrical potential, and zero-flux Neumann boundary conditions for the concentrations.

neurotransmitter dynamics, see, e.g., [24]. The PNP equations themselves are not restricted to the idealized cuboid geometry applied here, but more realistic synaptic cleft geometries would require a numerical method suitable for complex domains, such as the finite element method. Such extensions are possible, but are outside the scope of the present study.

Furthermore, the model does not include fixed surface charges associated with lipid head groups in the membrane bilayer. Such charges may influence the local electric potential and ionic concentrations close to the membrane. Representing this effect would require additional assumptions about membrane composition and surface charge density, and it is therefore left outside the present model. In addition, only a subset of the relevant ion transport is modeled: we do not include the arrival of the action potential in the presynaptic neuron or the possible generation of an action potential in the postsynaptic neuron. Moreover, the simulations cover a single vesicle release event. Extending the model to repeated firing would allow the study of concentration dynamics over longer time scales, including the accumulation or depletion of ionic species that may result from sustained synaptic activity.

Another limitation of the present approach is that PNP is a continuum, mean-field model, whereas vesicle release involves a limited number of glutamate molecules. At sufficiently fine spatial resolution, some local volumes would contain zero or one glutamate molecule, and the interpretation of a smooth concentration field therefore becomes less direct. The PNP solution should therefore be interpreted as an averaged description, rather than as an instantaneous particle realization. Particle-based stochastic or molecular dynamics models may be more appropriate when fluctuations in individual molecule positions are central to the question. However, to be comparable with the present model, such models would need to include five ion species present in the cleft, which would create a substantial computational challenge.

### 3.5. Conclusion

We have solved the full Poisson–Nernst–Planck (PNP) system in a three-dimensional model of the synaptic cleft and compared the results with those of the pure diffusion (D) model. The two models produce substantially different ionic concentration fields: the diffusive and electrical flux contributions are of comparable magnitude for all ionic species, and omitting the electrical term leads to errors that are robust across a wide range of parameter variations. These results establish that accurate computation of ionic concentrations in the synaptic cleft requires the full PNP equations. The model and numerical scheme introduced here are not specific to the particular process considered, and can be applied to study other aspects of the ionic dynamics in the synaptic cleft.

## 4. Methods

We study synaptic dynamics by considering a small part of a presynaptic cell, a small part of a postsynaptic cell, and the surrounding extracellular space (see Fig 1). In this section, the models and parameter values applied in our simulations will be described.

### 4.1. The Poisson–Nernst–Planck (PNP) model

The Poisson–Nernst–Planck (PNP) model for the electrical potential and ionic concentrations is given by the system [25]

$$\nabla \cdot \left( \varepsilon_r \varepsilon_0 \nabla \phi \right) = -\rho, \tag{5}$$

$$\frac{\partial [k]}{\partial t} = \nabla \cdot D_k \nabla [k] + \nabla \cdot \left( \frac{D_k z_k e}{k_B T} [k] \nabla \phi \right), \tag{6}$$

$$\rho = F \sum_k z_k [k], \tag{7}$$

for $k$={$Na^+$, $K^+$, $Ca^{2+}$, $Cl^-$, $Glu^-$}, representing sodium, potassium, calcium, chloride, and glutamate ions. Here, $\phi$ is the electric potential (in mV), $[k]$ is the concentration of ion species $k$ (in mM), and $\rho$ is the charge density (in C/m³). Furthermore, $\varepsilon_r$, $\varepsilon_0$, $D_k$, $z_k$, $e$, $k_B$, $T$, and $F$ are physical constants and parameters, specified in Table 1.

### 4.2. The pure diffusion (D) model

We compare the solution of the full PNP model (5)–(7) to the solution of the pure diffusion (D) model. In that model, flow of ions is assumed to be independent of electrical drift, and the ionic concentrations are governed by [25]

$$\frac{\partial [k]}{\partial t} = \nabla \cdot D_k \nabla [k],$$

(8)

for $k$={$Na^+$, $K^+$, $Ca^{2+}$, $Cl^-$, $Glu^-$}.

### 4.3. Parameter values, initial conditions and boundary conditions

The default parameter values used in our simulations are given in Table 1. The narrow extracellular cleft between the cells is assumed to have restricted diffusion, consistent with, e.g., [19,29]. More specifically, the diffusion coefficients in the synaptic cleft are divided by a scaling factor $\kappa$ (see Table 1). The initial conditions for the ionic concentrations in the different parts on the domain are reported in Table 2. We apply Dirichlet boundary conditions for both the concentrations and $\phi$ on the parts of the domain boundary that are denoted by $\partial\Omega_{e,d}$ or $\partial\Omega_{i,d}$ in Fig 1B. At $\partial\Omega_{e,d}$, $\phi$ is set to 0 mV and the concentrations are set to the extracellular concentrations given in Table 2. At $\partial\Omega_{i,d}$, $\phi$ is set to −70 mV, and the concentrations are set to the intracellular concentrations given in Table 2. On the remaining boundaries, homogeneous Neumann boundary conditions are applied for both $\phi$ and the concentrations.

### 4.4. Synaptic vesicle

The synaptic vesicle is represented as a volume in the presynaptic cell close to the synaptic cleft (see Fig 1). Initially, the vesicle is assumed to contain an electroneutral mix of negatively charged glutamate ions and positively charged $Na^+$ and $K^+$ ions (see Table 2). This choice is consistent with vesicle-loading models that keep luminal $Na^+$ and $K^+$ fixed at extracellular concentrations while glutamate accumulates [32]. As the simulation starts, ions are allowed to flow between the vesicle and the synaptic cleft through an opening of size $w_{vo}^2$ (see Fig 1 and Table 1).

### 4.5. AMPA receptor flux

We model the flux through the AMPA receptors using internal boundary conditions on the boundaries between the membrane and the intracellular and extracellular domains, as explained in detail in [14]. The AMPA receptors are assumed to be evenly distributed in the AMPA receptor area of the postsynaptic membrane (see Fig 1). The open probability of the

**Table 2. Initial conditions for the ionic concentrations in the intracellular and extracellular domains. The values are based on [7,17,28,32]. Some concentration values are reported with extra digits to ensure initial electroneutrality of the intracellular and extracellular solutions.**

| Ion | Extracellular | Intracellular | Synaptic vesicle |
|---|---|---|---|
| $Na^+$ | 100 mM | 12 mM | 145 mM |
| $K^+$ | 4 mM | 125 mM | 5 mM |
| $Ca^{2+}$ | 1.4 mM | 0.0001 mM | 0 mM |
| $Cl^-$ | 106.8 mM | 137.0002 mM | 0 mM |
| $Glu^-$ | 0 mM | 0 mM | 150 mM |

receptors depends on the local concentration of glutamate on the extracellular side of the receptors, represented using the Markov model from [33]. Furthermore, the AMPA receptor channels are assumed to be Na$^+$ and K$^+$ selective.

The Na$^+$ and K$^+$ flux densities through the AMPA receptors are given by

$$J_{\text{Na}^+} = \frac{N_{\text{AMPA}}}{Fw_a^2} g_{\text{AMPA,Na}^+} o(v - v_{0,\text{Na}^+}),$$

(9)

$$J_{\text{K}^+} = \frac{N_{\text{AMPA}}}{Fw_a^2} g_{\text{AMPA,K}^+} o(v - v_{0,\text{K}^+}),$$

(10)

respectively. Here $F$ is Faraday's constant, $N_{\text{AMPA}}$ is the number of AMPA receptors, $w_a$ is the width of the AMPA receptor area of the postsynaptic membrane. Moreover, $o$ is the open probability of the receptors (see below), $v = \phi_i - \phi_e$ is the potential difference across the membrane, and $v_{0,\text{Na}^+}$ and $v_{0,\text{K}^+}$ are the Nernst potentials of Na$^+$ and K$^+$, respectively. Furthermore, $g_{\text{AMPA,Na}^+}$ and $g_{\text{AMPA,K}^+}$ are the single channel AMPA receptor conductances for Na$^+$ and K$^+$, respectively. The values of $o$, $v$, $v_{0,\text{Na}^+}$ and $v_{0,\text{K}^+}$ are evaluated and applied locally for each mesh element in the AMPA receptor area. For the diffusion model, $v$ is set to −70 mV, corresponding to the holding potential used in the PNP system. In other words, the intracellular potential is assumed to be constantly equal to −70 mV and the extracellular potential is assumed to be constantly equal to 0 mV.

The corresponding AMPA receptor current is given by

$$I_{\text{AMPA}} = \int_{A_{\text{AMPA}}} F\left(J_{\text{Na}^+} + J_{\text{K}^+}\right) ds,$$

(11)

where $A_{\text{AMPA}}$ is the AMPA receptor area. Inserting (9) and (10), we get

$$I_{\text{AMPA}} = \frac{N_{\text{AMPA}}}{w_a^2} \int_{A_{\text{AMPA}}} o\left(g_{\text{AMPA,Na}^+}(v - v_{0,\text{Na}^+}) + g_{\text{AMPA,K}^+}(v - v_{0,\text{K}^+})\right) ds.$$

(12)

**4.5.1. Markov model for the AMPA receptor open probability.** The model for the AMPA receptor open probability is given by the following Markov model from [33]:

$$C_0 \underset{k_{-1}}{\overset{k_{+1}[\text{Glu}^-]}{\rightleftarrows}} C_1 \underset{k_{-2}}{\overset{k_{+2}[\text{Glu}^-]}{\rightleftarrows}} C_2 \underset{\beta}{\overset{\alpha}{\rightleftarrows}} O$$

with vertical transitions $\alpha_1 | \beta_1$, $\alpha_2 | \beta_2$, $\alpha_3 | \beta_3$

$$C_3 \underset{k_{-3}}{\overset{k_{+3}[\text{Glu}^-]}{\rightleftarrows}} C_4 \underset{\beta_4}{\overset{\alpha_4}{\rightleftarrows}} C_5$$

The ODE version of the Markov model reads,

$$\frac{d[C_0]}{dt} = -k_{+1}[\text{Glu}^-][C_0] + k_{-1}[C_1],$$

$$\frac{d[C_1]}{dt} = k_{+1}[\text{Glu}^-][C_0] - (k_{-1} + k_{+2}[\text{Glu}^-] + \alpha_1)[C_1] + k_{-2}[C_2] + \beta_1[C_3],$$

$$\frac{d[C_2]}{dt} = k_{+2}[\text{Glu}^-][C_1] - (k_{-2} + \alpha + \alpha_2)[C_2] + \beta[O] + \beta_2[C_4],$$

$$\frac{d[C_3]}{dt} = \alpha_1[C_1] - (\beta_1 + k_{+3}[\text{Glu}^-])[C_3] + k_{-3}[C_4],$$

$$\frac{d[C_4]}{dt} = k_{+3}[\text{Glu}^-][C_3] + \alpha_2[C_2] - (k_{-3} + \beta_2 + \alpha_4)[C_4] + \beta_4[C_5],$$

$$\frac{d[C_5]}{dt} = \alpha_3[O] + \alpha_4[C_4] - (\beta_3 + \beta_4)[C_5],$$

$$[O] = 1 - ([C_1] + [C_2] + [C_3] + [C_4] + [C_5]),$$

where $[\text{Glu}^-]$ is the local glutamate concentration (in mM), $[X]$ is the probability of being in state X, and $k_{+1}$, $k_{-1}$, $k_{+2}$, $k_{-2}$, $k_{+3}$, $k_{-3}$, $\alpha$, $\beta$, $\alpha_1$, $\beta_1$, $\alpha_2$, $\beta_2$, $\alpha_3$, $\beta_3$, $\alpha_4$, $\beta_4$ are parameter values given in Table 3. The AMPA open probability is defined as $o = [O]$.

### 4.6. Numerical methods

The PNP and D equations are solved numerically using the finite difference method with an adaptive spatial mesh (see, e.g., [13]). We generally use a time step of 0.02 ms, except in the simulation used to display the dynamics in the beginning of the simulation in Fig 2, where a time step of 0.001 ms is applied. The D equations are solved using a standard implicit Euler method, and the PNP equations are solved in a coupled manner using the scheme from [14,22]. The AMPA receptor open probability model is handled separately using a standard operator splitting technique (see, e.g., [25]) with a local time step of 0.001 ms.

**Table 3. Parameter values for the Markov model for the AMPA receptor kinetics. The model and parameters are taken from [33].**

| Parameter | Value | Parameter | Value |
|---|---|---|---|
| $k_{+1}$ | $4.59 \text{ mM}^{-1}ms^{-1}$ | $k_{-1}$ | $4.26 \text{ ms}^{-1}$ |
| $k_{+2}$ | $28.4 \text{ mM}^{-1}ms^{-1}$ | $k_{-2}$ | $3.26 \text{ ms}^{-1}$ |
| $k_{+3}$ | $1.27 \text{ mM}^{-1}ms^{-1}$ | $k_{-3}$ | $0.0457 \text{ ms}^{-1}$ |
| $\alpha$ | $4.24 \text{ ms}^{-1}$ | $\beta$ | $0.9 \text{ ms}^{-1}$ |
| $\alpha_1$ | $2.89 \text{ ms}^{-1}$ | $\beta_1$ | $0.0392 \text{ ms}^{-1}$ |
| $\alpha_2$ | $0.172 \text{ ms}^{-1}$ | $\beta_2$ | $7.27 \cdot 10^{-4} \text{ ms}^{-1}$ |
| $\alpha_3$ | $0.0177 \text{ ms}^{-1}$ | $\beta_3$ | $4.0 \cdot 10^{-3} \text{ ms}^{-1}$ |
| $\alpha_4$ | $0.0168 \text{ ms}^{-1}$ | $\beta_4$ | $0.1904 \text{ ms}^{-1}$ |

## Supporting information

**S1 Fig: Glutamate dynamics displayed with equal scaling of the axes.** The solutions are the same as those displayed in the upper panel of Fig 2, but in Fig 2 the y-axis is stretched to make the solutions more compact.
(TIF)

## Acknowledgments

Disclosure of writing assistance: During the preparation of this manuscript, the authors utilized the ChatGPT5.1 language model to enhance the language quality. Subsequent to this automated assistance, the authors rigorously reviewed and edited the manuscript to ensure its accuracy and integrity. The authors assume full responsibility for the content of the publication.

## Author contributions

**Conceptualization:** Karoline Horgmo Jæger, Aslak Tveito.

**Formal analysis:** Karoline Horgmo Jæger.

**Funding acquisition:** Karoline Horgmo Jæger, Aslak Tveito.

**Investigation:** Karoline Horgmo Jæger, Aslak Tveito.

**Methodology:** Karoline Horgmo Jæger, Aslak Tveito.

**Software:** Karoline Horgmo Jæger.

**Visualization:** Karoline Horgmo Jæger, Aslak Tveito.

**Writing – original draft:** Karoline Horgmo Jæger, Aslak Tveito.

**Writing – review & editing:** Karoline Horgmo Jæger, Aslak Tveito.

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
