## [Decision Letter · Decision Letter 0]

23 Apr 2026

PCOMPBIOL-D-26-00563

Accurate computation of ionic concentrations in the synaptic cleft requires the full Poisson–Nernst–Planck (PNP) equations

PLOS Computational Biology

Dear Dr. Jæger,

Thank you for submitting your manuscript to PLOS Computational Biology. After careful consideration, we feel that it has merit but does not fully meet PLOS Computational Biology's publication criteria as it currently stands. Therefore, we invite you to submit a revised version of the manuscript that addresses the points raised during the review process.

We look forward to receiving your revised manuscript.

Kind regards,

Alain Nogaret, PhD

Academic Editor

PLOS Computational Biology

Daniele Marinazzo

Section Editor

PLOS Computational Biology

**Additional Editor Comments:**

You will see reviewers support publication of your work but have prior requested a number of mandatory revisions to be implemented. Please carefully address this comments in the revised manuscript and your response.

**Journal Requirements:**

2) Your manuscript's sections are not in the correct order.  Please amend to the following order: Abstract, Introduction, Results, Discussion, and Methods

4) In the online submission form, you indicated that your data will be submitted to a repository upon acceptance. We strongly recommend all authors deposit their data before acceptance, as the process can be lengthy and hold up publication timelines. Please note that, though access restrictions are acceptable now, your entire minimal dataset will need to be made freely accessible if your manuscript is accepted for publication. This policy applies to all data except where public deposition would breach compliance with the protocol approved by your research ethics board. If you are unable to adhere to our open data policy, please kindly revise your statement to explain your reasoning and we will seek the editor's input on an exemption.

**Reviewers' comments:**

Reviewer's Responses to Questions

**Comments to the Authors:**

Reviewer #1: This is a nice paper showing the significant consequences of neglecting electrical field effects on ion diffusion in a synaptic cleft of a glutamatergic AMPA synapse. The authors compare purely diffusive model with the full Poisson-Nerst-Planck model of coupled diffusion of individual ionic species with the Poisson equation on the electric potential.

The paper is well written, the models and simulations are clearly defined and described. Results are adequately illustrated, even if the individual figure panels are rather small, I think they are sufficient to represent the individual effects that are observed and discussed.

I have not seen errors. Some minor comments:

- I would finish the first sentence of the second paragraph in the Introduction with "for glutamatergic AMPA synapse" etc., I had a moment of hesitation on the generality of the announced research. I understand the potential for translation, of course.

- in Table 1 and 2 do we need so many significant digits? In particular, how valid is providing the concentration of intracellular Cloride with 7 significant digits?

- In 3.4.3 "fluxes into or out of the cleft has" -> have?

- In 3.4.4 "fluxes into or out of the cleft causes" -> cause?

Since the main point of this paper is that the effects of the electric field are often comparable in size to the effects of diffusion, so the PNP is more realistic, I have a question on the realism of the model, specifically, how realistic is the continuous description of this process.

Given that a single synaptic vesicle will have several thousands glutamate molecules, and given the spatial precision of the figures, consider coarse-graining on the order of 10um or 1um per side. For cleft of 20um that would give us a partition into 7200 boxes. For a finer partition of 1um we would have 7.2 mln. boxes. That would give us very sparse occupation with Glutamate particles. So I wonder if molecular dynamics in the end is not more adequate in this context? I realize a comparison of PNP with MD simulations is a different paper, but I think the paper would benefit from a brief discussion of the effects of low particle numbers in the cleft on the realism of those results and possible relation to MD simulations.

Unfortunately, the code for the paper is not provided so I was not able to inspect it. Making the code available would increase the impact of this work in the community. I encourage the authors to make their code public in a way which would encourage reproduction of those results.

According to the submission the code is available on Zenodo (DOI: 10.5281/zenodo.18952935). I was unable to find it.

Reviewer #2: The authors explore simulating the release of neurotransmitter from a single vesicle into an idealized synaptic cleft and the response of AMPA receptors in the postsynaptic membrane. They compare Poisson-Nernst-Plank (PNP) and pure diffusion models and conclude that inclusion of the electrical potentials in the PNP model generates significantly different predictions than the diffusion model which ignores electrical effects. This could have important ramifications for modeling synaptic neurotransmission, and thus is an important observation. As such, I am mostly enthusiastic about this work. There are a few points that need to be addressed prior to publication (see comments below for details). Primarily, they involve using a more realistic Markov model for AMPA receptors (this is a must), explaining what appears to be odd diffusional bias along certain dimensions, and reporting the comparison of PNP and diffusion models on the simulated postsynaptic current.

1. The Markov model for AMPA receptors has a serious flaw in that it does not include desensitization. AMPA receptor currents are shaped by rapid activation and also rapid desensitization, which must be included for any realistic model of AMPA receptor behavior. This could also influence the final conclusion as to how much the PNP vs Diffusion models actually differ in their predictions for postsynaptic currents.

2. In Fig. 2, why does Glu appear to diffuse/disperse primarily in the y-direction and not z-direction? This seems odd to me. The depicted Glu transient appears almost like a laser that zaps across the synaptic cleft without much lateral diffusion. Can this be reconciled with spillover of neurotransmitter outside of the synaptic cleft which we know occurs? Some discussion as to what is driving this behavior should be included. The current explanation of this behavior simply describes the events without providing much explanation as to why they occur. Also, although there is more diffusion in the z-direction for the diffusion model, both Glu and ions still appear to primarily diffuse in the y-direction (see Fig. 3). Does this make sense? For the diffusion model, why should there be any bias for diffusion in y- vs z-directions, or the y- vs x- directions for that matter? As shown, even the diffusion model appears to predict a broader laser zapping across the cleft. It makes me worry that there is something wrong with the simulation, although perhaps I am missing something. Either way, an explanation would be very helpful for most readers (and myself).

3. Ultimately, what matters most for those interested in simulating synaptic communication is the postsynaptic response. Although the authors thorough description of the various in-cleft distributions of Glu and ions is certainly still highly relevant, I would ask them to report the simulated postsynaptic current through AMPA receptors, i.e., the EPSC, and to compare the predicted EPSC between PNP and diffusion models. Of course, this must be done with a physiologically realistic AMPA receptor Markov model that includes desensitization. This information is crucial for many to decide whether the added cost of the PNP model is worth it for their purposes.

4. There are charges on the lipid head groups in the membrane bilayer. It is unclear to what extent this is accounted for or if it matters.

5. It is unclear how easily the PNP solution can be extended to more realistic synaptic cleft geometries. A brief comment on this would be helpful.

**Have the authors made all data and (if applicable) computational code underlying the findings in their manuscript fully available?**

Reviewer #1: **No:** The code has not been provided. I encourage the authors to make their code public in a way which would encourage reproduction of those results.

Reviewer #2: Yes

PLOS authors have the option to publish the peer review history of their article (what does this mean?). If published, this will include your full peer review and any attached files.

**Do you want your identity to be public for this peer review?** For information about this choice, including consent withdrawal, please see our Privacy Policy.

Reviewer #1: No

Reviewer #2: **Yes:** Marcel Goldschen-Ohm

**Figure resubmission:**
---

## [Decision Letter · Decision Letter 1]

18 May 2026

Dear Dr. Jæger,

We are pleased to inform you that your manuscript 'Accurate computation of ionic concentrations in the synaptic cleft requires the full Poisson–Nernst–Planck (PNP) equations' has been provisionally accepted for publication in PLOS Computational Biology.

Best regards,

Alain Nogaret, PhD

Academic Editor

PLOS Computational Biology

Daniele Marinazzo

Section Editor

PLOS Computational Biology

The manuscript is now suitable for publication.

Reviewer's Responses to Questions

**Comments to the Authors:**

Reviewer #2: The authors have addressed all of my comments.

**Have the authors made all data and (if applicable) computational code underlying the findings in their manuscript fully available?**

Reviewer #2: Yes

PLOS authors have the option to publish the peer review history of their article (what does this mean?). If published, this will include your full peer review and any attached files.

Reviewer #2: **Yes:** Marcel Goldschen-Ohm

---

## [Editor Report · Acceptance letter]

PCOMPBIOL-D-26-00563R1

Accurate computation of ionic concentrations in the synaptic cleft requires the full Poisson–Nernst–Planck (PNP) equations

Dear Dr Jæger,

I am pleased to inform you that your manuscript has been formally accepted for publication in PLOS Computational Biology. Your manuscript is now with our production department and you will be notified of the publication date in due course.

With kind regards,

Judit Kozma
